# An Overview of Millimeter-Wave Radar Modeling Methods for Autonomous Driving Simulation Applications

**DOI:** 10.3390/s24113310

**Published:** 2024-05-22

**Authors:** Kaibo Huang, Juan Ding, Weiwen Deng

**Affiliations:** 1School of Transportation Science and Engineering, Beihang University, Beijing 100191, China; wdeng@buaa.edu.cn; 2PanoSim Technology Limited Company, Jiaxing 314000, China; juan.ding@panosim.com

**Keywords:** millimeter-wave radar, radar modeling, autonomous driving, ray tracing, machine learning

## Abstract

Autonomous driving technology is considered the trend of future transportation. Millimeter-wave radar, with its ability for long-distance detection and all-weather operation, is a key sensor for autonomous driving. The development of various technologies in autonomous driving relies on extensive simulation testing, wherein simulating the output of real radar through radar models plays a crucial role. Currently, there are numerous distinctive radar modeling methods. To facilitate the better application and development of radar modeling methods, this study first analyzes the mechanism of radar detection and the interference factors it faces, to clarify the content of modeling and the key factors influencing modeling quality. Then, based on the actual application requirements, key indicators for measuring radar model performance are proposed. Furthermore, a comprehensive introduction is provided to various radar modeling techniques, along with the principles and relevant research progress. The advantages and disadvantages of these modeling methods are evaluated to determine their characteristics. Lastly, considering the development trends of autonomous driving technology, the future direction of radar modeling techniques is analyzed. Through the above content, this paper provides useful references and assistance for the development and application of radar modeling methods.

## 1. Introduction

Statistical data reveal that about 1.3 million people die in road traffic accidents globally each year [1], with 94% of serious accidents being caused by human error. Autonomous driving is regarded as one of the pivotal technologies that will influence and reshape future traffic. Mature autonomous driving can effectively reduce traffic accidents, optimize traffic flow, and reduce energy consumption [2], possessing significant social value and broad market prospects. As an application of artificial intelligence, autonomous driving algorithm development necessitates extensive testing [3]. It is widely acknowledged in the industry that an autonomous driving algorithm requires at least 17 billion kilometers of test data to qualify for mass production [4]. Autonomous driving test data are derived from both simulation and road testing. The latter involves high costs and inefficiency in data collection and labeling, along with limited controllability and repeatability, making it challenging to acquire data in extreme scenarios. In virtual environments, it is possible to set up and record tests for any driving scenario, enabling the rapid generation of a vast array of scenario-specific test data, significantly cutting down both the time and financial costs associated with algorithm development testing. Based on these advantages, simulation technology has become a key tool in advancing the development of autonomous driving.

Autonomous driving technology is composed of four principal modules: perception, decision-making, planning, and control, with the perception module providing fundamental support for the other three. Autonomous driving employs sensors like millimeter-wave radar, cameras, lidar, ultrasonic radar, and inertial navigation systems for environmental perception, as depicted in Figure 1. Radar operates in the millimeter-wave band, situated between microwave and centimeter wavelengths, thereby possessing a detection scope and all-weather operation capability far surpassing other sensors [5,6,7]. Additionally, millimeter-wave radar employs Frequency-Modulated Continuous-Wave (FMCW) detection, offering capabilities for high-precision ranging and velocity measurement. With advancements in higher resolution [8] and full polarization technologies [9], radar has become an essential sensor within autonomous driving [10,11,12,13].

Autonomous driving simulation systems employ radar models to replicate the behavior of real-world radars. Radar models play a crucial role across multiple stages of autonomous driving development. As depicted in Figure 2, radar development teams employ radar models to facilitate the development of radar radio frequency (RF) chips and algorithms; autonomous driving solution providers use radar models for simulation testing of radar selection and installation locations; algorithm developers rely on radar models to develop sensor fusion and decision-planning algorithms, and data-driven methods require generating a large amount of simulation data for data augmentation. The perception outcomes of radar models directly influence the realism and reliability of autonomous driving simulation systems. High-performance radar models are crucial components within autonomous driving simulation systems [14].

Current overviews of radar modeling methods emphasize the practical applications of radar models in autonomous driving systems [15], offering limited insight into radar working mechanisms and a lack of analysis on content and critical factors impacting model efficacy. This makes it difficult to identify the keys and challenges affecting radar modeling effectiveness. Moreover, current radar modeling evaluation standards are overly simplified [16] and do not adequately distinguish the specific requirements for radar models across different application levels. This paper extensively examines the working principles of radar and the interference factors encountered in detection, further clarifying the core contents of radar modeling and the key factors that influence it. From the perspective of practical applications, it specifies the performance demands of radar models at different levels, establishing clear evaluation criteria for radar models. Then, it introduces modeling methods from radar sub-modules to the overall system, focusing on analyzing the principles, current research progress, and advantages and disadvantages of various modeling methods, and evaluates each modeling method from a practical application perspective. Finally, this paper predicts the future development direction of radar models based on the direction of the development of autonomous driving.

The second part of this paper illustrates the contents of radar modeling by introducing the working process of radar and the influencing factors in its operation. The third part of the article clarifies the evaluation indicators of radar models by introducing the requirements of different applications for radar models. The fourth part of the article evaluates existing radar modeling methods by introducing their principles, elucidating the characteristics of each modeling method and the completeness of modeling content, and evaluating them according to evaluation indicators. The fifth part of the article analyzes the future development direction of radar modeling methods based on the development direction of future autonomous driving and the characteristics of existing radar modeling methods. The sixth part is the summary of the entire paper.

## 2. Principles of Autonomous Driving Radar

The radar system is a complex multi-module system, covering various aspects such as radio frequency technology, electromagnetic wave propagation in space, and radar algorithms. To construct an accurate radar model, it is essential to clearly define the content and workflow of the radar system and the main factors influencing its performance.

### 2.1. The Content and Workflow of Radar System

As shown in Figure 3, the radar system can be divided into three modules based on the working mechanism differences: the radar functional module, representing the main features of the radar, includes the antenna array and RF circuits and generates radar signals and handles the reception, transmission, and RF signal processing to obtain intermediate frequency signals with information about detected targets; the second module is the echo module, which includes scene information, as well as the entire process of radar emitting electromagnetic waves, electromagnetic wave propagation in space, and generating echo signals; the last module is the radar algorithm module, which analyzes digital signals containing target information, identifying targets and extracting their motion parameters [17,18].

The RF hardware of the radar includes two submodules: the transmitter and the receiver. The complete workflow of the radar system is as follows. The transmitter first sets the waveform parameters in the waveform generator; then, the voltage-controlled oscillator (VCO) and phase-locked loop (PLL) generate the modulated signal of that waveform. Subsequently, the signal is amplified to a frequency near 77 GHz by a frequency multiplier, then divided by a power divider, with one part of the signal transmitted to the mixer in the receiving module. Another path transmits the signal to the power amplifier (PA) in the transmitting module, where it is amplified before being emitted through the transmitting antenna array (TX). The emitted signal freely travels through space and reflects off objects, with part of the signal returning to the radar’s receiving antenna array (RX) as the received echo signal. The radiation patterns of the transmit and receive antenna arrays apply varying levels of gain to signals from different directions, thereby enhancing the radar’s detection capabilities towards the directions of interest. After receiving the echo signal, the radar receiver amplifies the signal by a low noise amplifier (LNA) and then sends it to the mixer for mixing with the transmitter’s signal to obtain the intermediate frequency signal. This signal is then filtered to remove high-frequency interference signals and converted into a digital signal by the analog-to-digital conversion (ADC) module. This signal, containing target information, is also known as raw data. The radar’s digital signal processing and algorithm module then perform three-dimensional fast Fourier transform (3D-FFT) processing on the incoming raw data to extract information on range, velocity, and angle, generating two key echo maps: the range-angle map (RAmap), containing range and angle information, and the range-Doppler map (RDmap), containing range and velocity information. These two maps are then used in the constant false alarm rate (CFAR) algorithm to identify target point clouds, which are subsequently clustered using a clustering algorithm to output identified targets. Finally, a target tracking algorithm is used to track and correct targets, outputting a list of targets along with their corresponding parameters [19,20,21,22]. Radar detection results encompass parameters like target energy intensity, range, angle, and velocity, typically represented by parameters such as RCS, R, V, and θ. RCS represents the energy intensity of the target signal received by the radar, measured in dBm. R signifies the range to the target from the radar, in meters. V indicates the target’s relative radial velocity to the radar, in meters per second (m/s). θ  denotes the angle between the target and the radar’s normal, in degrees.

### 2.2. Interference Factors in Radar System

In an ideal setting, radar can accurately identify targets within a scene and detect their real motion information. However, in practical applications, the detection process of radar is affected by various interference factors, resulting in multiple errors during radar detection.

In the radar transmitter, the VCO and PLL are usually subject to limitations in component performance, leading to issues such as frequency offset and phase noise. Particularly during frequency modulation, if the phase-locked loop fails to fully converge, signal frequency may fluctuate [23,24]. Frequency multipliers and power dividers have a minor impact on the signal transmission efficiency and the power of the output signal. During signal amplification, PAs not only produce thermal noise but also exhibit gain change with increasing input signal power.

In the receiver, LNAs face challenges such as thermal noise and gain nonlinearity, which have a greater impact on radar signals than the amplifiers in the radar transmitter. The non-ideal transmission coefficient of filters may result in incomplete removal of interference signals. Another crucial component that significantly affects signals is the ADC, whose internal quantization noise and phase noise may introduce additional interference to the signal.

The echo signals received by radar are often influenced by factors such as clutter and multipath in free space. Clutter includes ground clutter and weather clutter: ground clutter refers to electromagnetic signals reflected by non-target objects such as the ground, buildings, and trees, while weather clutter refers to electromagnetic signals scattered by water vapor particles such as raindrops, snowflakes, and fog in the atmosphere. These clutter signals mix with the reflected signals from scene targets, causing interference to the echo signals. Additionally, weather factors may cause amplitude attenuation and phase changes in target echo signals. Multipath signals are generated by electromagnetic signals undergoing multiple reflections, refractions, diffractions, or scatterings between targets, ground, or buildings in the scene. These signals carry erroneous target information and, when mixed with the echo signals from real targets, interfere with the radar’s reception of real target signals.

Three-dimensional-FFT directly influences radar resolution. In the distance dimension, resolution is closely related to the bandwidth of the FMCW signal; the larger the bandwidth, the higher the distance resolution. Velocity resolution depends on the observation time of the radar and the period of the chirp. The more pulses observed within the observation time, the higher the velocity measurement accuracy. The more pulses observed within the observation time, the higher the velocity measurement accuracy. Angle resolution is affected by the design of the antenna array and the number of antennas. The greater the number of antennas, the higher the angle resolution.

In radar algorithms, the CFAR algorithm in cluttered environments may incorrectly identify noise or non-target echoes as targets. Moreover, when the target’s reflected intensity approaches the noise level, CFAR may fail to detect real targets correctly. Improperly designed clustering algorithms may lead to an inability to accurately distinguish neighboring targets or to inaccurate estimates of target size, shape, and position, thereby affecting target classification and tracking. Poorly designed tracking modules in radar may result in target loss or inaccurate tracking, thereby affecting the radar system’s continuous monitoring and prediction of targets. Additionally, due to the limited resolution of radar, the point cloud data output by CFAR may be sparse. Sparse point clouds may lead to inaccurate target detection and tracking, especially in dynamic environments where sparse point clouds may not provide sufficient information to accurately estimate the motion state of targets.

In the presence of various interference factors, radar outputs may include missed alarms, false alarms, and measurement errors. Missed alarms refer to the scenario where targets within the scene are not identified. Normally, the echo signal strength from targets in the scene is high enough for the radar to detect the echo signal reflected back by the target from the signal containing noise and other interferences, thereby identifying the target, as shown in Figure 4a. When there is obstruction between targets in space, the target is too far from the radar, or the interference signal strength is too high, the echo signal strength of the target significantly decreases and radar algorithms cannot identify targets from interference signals, leading to missed alarms, as shown in Figure 4b.

False alarms refer to the radar identifying targets that do not exist in the scene, also known as ghost objects. False alarms are often the result of strong reflections from large structures, such as the ground or buildings, and from metal targets. When specific geometric relationships exist between the radar and these reflective surfaces, the multipath signals between them are received by the radar. As shown in Figure 5, these multipath signals, due to their high intensity, are not filtered out by the radar’s algorithms. Instead, they are mistakenly recognized as legitimate objects.

Ranging errors in radar arise from multiple factors. With a fixed radar FMCW cycle, the modulation bandwidth and radar ranging error have an inverse relationship (1), where c is the speed of light and B represents the bandwidth of FMCW. Changes in the linearity of FMCW modulation $l_{d}$ also affect the mixed signal, causing range errors, and the magnitude of range errors is directly proportional to the actual range between the target and the radar (2); when the target has radial motion relative to the radar, the Doppler frequency shift produced by the radial velocity V also interferes with the radar’s ranging results (3), where T represents the frequency modulation period of FMCW, λ represents the wavelength corresponding to the operating frequency, and θ represents the angle between the target and the radar normal [25,26,27].
(1)$$\Delta R_{1_{\;}}=\frac{c}{2B}\;,$$
(2)$$\Delta R_{2}=l_{d}\cdot R\;,$$
(3)$$\Delta R_{3}=\frac{c{TV}_{\;}cos\theta }{B\lambda }\;,$$

In terms of radar velocity measurement, the received signal is a superposition of the reflected echoes from various parts of the target, with different velocities at different parts resulting in varying Doppler frequency shifts. Thus, the calculated radar velocity will have a certain deviation (4), where ω is the angular velocity of the target’s relative motion, *W* is the target’s lateral length, and *f* is the radar operating frequency. Additionally, when calculating target parameters, signals are usually accumulated first to increase the signal-to-noise ratio of the echo signal. When there is relative acceleration between the target and the radar, the signal frequencies at different times vary, leading to velocity measurement errors (5), where *a* is the target’s angular acceleration, and $T_{a}$ is the signal accumulation time. Noise and clutter contained in the radar signal are another cause of velocity measurement errors (6), where N is the power of noise clutter, and S is the total signal power. A deviation between the radar’s actual frequency modulation slope and the ideal slope, especially if the transmitter’s frequency modulation linearity is poor, can also lead to velocity measurement errors (7) [28,29].
(4)$$\Delta V_{1}=\frac{2\omega Wf_{\;}}{c},$$
(5)$$\Delta V_{2}=\frac{2a_{\;}T_{a}f_{\;}}{c},$$
(6)$$\Delta V_{3}=\frac{\sqrt{T_{\;}B}}{T_{a}\sqrt{S/N}}\;,$$
(7)$$\Delta V_{4}=l_{d}\cdot V,$$

Angle measurement errors are determined by the accuracy of radar phase measurement, which is related to the distance d between antennas in the radar receiving array, the working wavelength, and the angle θ of the target [30].
(8)$$\Delta \theta =\frac{\lambda }{2\pi dcos\theta }d\varphi ,$$

The radar-detected target *RCS*, as seen from (9), is influenced by multiple parameters. The signal integrates external interference and internal noise, which are the main factors causing *RCS* detection errors. Here, $P_{r}$ is the power of the received signal, $P_{0}$ is the power of the transmitted signal, and $G_{i}$ and $G_{r}$ are the gains of the transmitting and receiving antennas, respectively.
(9)$$RCS=\frac{(4\pi {)}^{3}R^{4}P_{r}}{P_{0}G_{i}G_{r}\lambda^{2}},$$

## 3. Radar Model Evaluation Methods

Radar modeling methods have their own characteristics. To select the appropriate method suitable for different applications, a reasonable evaluation method is needed to assess existing radar modeling methods. Currently, the common evaluation method classifies models into low-fidelity models, medium-fidelity models, and high-fidelity models based on their accuracy. However, this method of evaluation is neither clear nor comprehensive enough. It is necessary to consider the practical application perspective and comprehensively evaluate radar models.

The prediction accuracy of radar models directly influences the credibility of simulation results, which is an important indicator. The accuracy of the model depends on its fitting various levels of content and influencing factors in the radar workflow. In the early development of automatic driving decision-making and planning algorithms, the accuracy requirement for radar models was relatively low, only needing to identify targets in the scene. Such radar models only need to include partial content of the radar workflow to be implemented and are referred to as low-fidelity models. With technological advancements, the accuracy requirements for radar models in various applications have gradually increased. For instance, data augmentation through simulation data to expand dataset capacity, radar selection, or simulation of radar installation positions all impose higher demands on radar simulation accuracy. These applications do not require radar models to be completely consistent with real radar outputs but require the radar model to accurately reflect the characteristics of real radar operation and avoid significant errors. Achieving this level of simulation accuracy requires radar models to accurately fit most of the content and some interfering factors in the radar workflow, and such models are referred to as medium-fidelity models. High-level applications such as the development of advanced automatic driving decision-making and planning algorithms, sensor fusion, radar algorithm development, or radar RF module development can only be effective and ensure simulation confidence when the accuracy of radar models is sufficiently high. Such applications particularly require radar models to reflect the influence of various interfering factors in real radar operation. Radar models that meet these application requirements should be able to fit almost all content and interfering factors in the radar workflow and are known as high-fidelity radar models.

Sensor fusion algorithms and decision planning algorithms process data from radar, cameras, and other sensors in real-time according to predefined time sequences. These applications require radar models to output detection results at a frequency consistent with the real radar detection cycle, which is typically in the millisecond range (50–100 ms). Therefore, the computational speed of radar models should match or exceed the radar detection cycle to ensure real-time outputs. Such radar models are termed high real-time radar models. Other applications, such as radar RF development, radar algorithm development, data augmentation, and radar selection, do not require strict real-time constraints on simulations. In these cases, the real-time performance of radar models is not a critical factor, and they are referred to as low real-time radar models.

The development of decision planning algorithms, target-level sensor fusion, target-level data augmentation, and radar selection necessitate radar models to output corresponding object lists based on scene information. Although these models cover the entire process of radar operation, they do not necessitate the internal processing of raw signal data. Instead, they are designed to directly generate the final object list, which categorizes them as object-level radar models. Applications such as radar algorithm development, signal-level sensor fusion, signal-level data augmentation, and radar installation position simulation require radar models to provide more raw signal data, termed signal-level models. These models typically output RAmap and RDmap processed by 3D-FFT, without the radar algorithm module after 3D-FFT. This paper defines target-level requirements as low openness requirements and signal-level requirements as high openness requirements.

One of the main objectives of autonomous driving simulation is to acquire data from various hazardous scenarios. In this scenario, there is a demand for the generalization capability of radar models, especially in applications such as decision planning algorithm development, sensor fusion, data augmentation, and radar algorithm development, where diverse scenarios are needed to ensure system robustness. We refer to this demand for radar models as high generalization requirements. In contrast, radar RF hardware development, radar installation position simulation, and radar selection typically simulate only typical scenarios, with lower requirements for the model’s generalization capability. We refer to this situation as low generalization requirements.

In summary, the evaluation metrics for a radar model depend on its simulation accuracy, speed, openness, and generalizability. The levels of demand for radar models by different applications are as shown in Table 1.

## 4. Radar Modeling Methods for Autonomous Driving

The working mechanisms of the three modules in the radar system differ significantly, and correspondingly, the modeling methods also vary greatly. This paper first introduces the modeling methods for the RF functional module, echo module, and radar algorithm module. Based on this, the overall modeling method of signal-level and object-level radar models is introduced, and the advantages and disadvantages of various modeling methods are analyzed. Finally, the performance of various modeling methods is evaluated.

### 4.1. Modeling Methods for Internal Modules

The radar model is subdivided into three parts: functional model, echo model, and algorithm model. The functional model includes the transmitter model and receiver model. The input of the functional model is the received echo signal, and the output is raw data. Typically, the functional modeling method involves creating models of radar RF components by formulating their functions based on the working mechanisms of the actual components. Subsequently, these individual RF component models are integrated to assemble a comprehensive functional model.

The input of the echo model consists of the electromagnetic wave information emitted by the functional model and the scene information, and its output is the echo signal received by the radar. It comprises four parts: the echo signal generated by the target, the echo signals caused by multipath effects, the clutter produced by ground, and the clutter and attenuation caused by weather factors. In the modeling methods for the echo model, the ideal mechanism method considers the targets in the scene as point targets and directly calculates the target echoes based on radar detection formulas. The ray tracing method is based on high-frequency methods such as geometric optics and physical optics to calculate the target and multipath-generated echo signals as well as ground clutter. Empirical formulas based on statistics are commonly used to calculate signal attenuation and clutter caused by weather factors, as well as ground clutter.

The algorithm model includes a 3D-FFT module for processing raw data and radar algorithms such as CFAR, clustering, and tracking. Its input is raw data, and the 3D-FFT module outputs signals like RAmap or RDmap, while the radar algorithms output an object list. Radar algorithms typically employ methods such as Cell Averaging CFAR (CA-CFAR), density-based spatial clustering of applications with noise (DBSCAN) clustering, and Kalman filtering.

#### 4.1.1. Echo Modeling Method Based on Ray Tracing

Ray tracing treats electromagnetic waves as rays emitted by the radar, which carry energy and propagate forward [31,32,33]. Upon encountering targets, these rays undergo reflection and scattering. The trajectory of these rays in space is calculated through reflection, and the scattered electromagnetic field at the radar’s receiving antenna is analyzed to deduce the radar’s received echo signal. The steps are as follows [34,35,36,37]: (1)Scene Construction: Establish a three-dimensional scene comprising various objects such as roads, buildings, vehicles, pedestrians, etc. Define the geometric shapes, positions, and material properties of each object, and partition the surfaces.(2)Ray Emission: Emit rays outward from the radar antenna position.(3)Ray Propagation: Calculate the propagation path of rays in the scene based on the principles of geometric optics, until reaching the maximum number of reflections or when the energy attenuates to a set threshold.(4)Echo Signal Calculation: For rays incident on surface elements, calculate the equivalent current on the surface elements based on the physical optics method, and integrate over-the-surface elements to obtain the radiation field of the current at the radar receiving antenna. The radiation fields of all incident surface elements are vectorially summed to obtain the echo signal received by the radar, including frequency, phase, amplitude, and time delay between receiving the echo and the ray emission time.

This method considers multiple reflections of rays in the scene, simulating multipath influence in actual scenarios. Additionally, by calculating the echo signal through scattered waves, ground clutter can be computed.

#### 4.1.2. Echo Modeling Method Based on Ideal Mechanisms

The ideal-mechanism-based echo modeling method is widely used in radar simulation in the field of autonomous driving to generate echo signals. These methods ignore fixed objects in the scene such as the ground and buildings. Targets such as vehicles or pedestrians are treated as point targets, disregarding the effects of their shape and material on the direction and intensity of electromagnetic wave transmission. Additionally, interference signals such as multipath are disregarded. This method simplifies the propagation process of electromagnetic waves in space as follows:(1)The radar emits electromagnetic waves to the equivalent point of the target’s position.(2)Echo signal calculation. The electromagnetic waves emitted by the radar are directly reflected back to the radar’s receiving antenna from the equivalent point of the target. Phase changes are not considered in the calculation of the echo signal, and the echo signal strength returned by the target is calculated based on the transmission formula of electromagnetic waves (10), where σ represents the theoretical RCS of the target. The Doppler frequency shift of the echo signal is then calculated based on the relative velocity between the target and the radar (11), and the signal’s time delay ∆t (12) is determined based on the distance between the target and the radar.
(10)$$P_{r}=\frac{\sigma P_{0}\lambda^{2}}{{\left(4\pi \right)}^{3}R^{4}},$$
(11)$$f_{D}=V\cdot \frac{f_{\;}}{c},$$
(12)$$∆t=\frac{2R}{c},$$

#### 4.1.3. Statistical-Based Environmental Impact Modeling Method

Impact of weather factors on radar echo signals cannot be obtained through mechanistic derivation and requires empirical formulas established from statistical data to simulate their effects. Among them, the signal attenuation caused by rainfall, $\gamma_{rain}$ (dB/km), can be expressed as (13), where M is the rainfall intensity (mm/h), τ is the polarization tilt angle, and $\;k_{H},\;\;\alpha_{H},\;\;k_{V},\;\;\alpha_{V}$ are regression coefficients.
(13)$$\gamma_{rain}=kM^{\alpha },$$
(14)$$k=\left[k_{H}+k_{V}+\left(k_{H}-k_{V}\right)cos2\tau \right]/2,$$
(15)$$\alpha =\left[k_{H}\alpha_{H}+k_{V}\alpha_{V}+\left(k_{H}\alpha_{H}-k_{V}\alpha_{V}\right)cos2\tau \right]/2k,$$

The backscatter cross-section $\eta_{rain}$ of rainfall per unit volume is obtained from (16), where D (mm) is the raindrop diameter, N(D)dD represents the number of raindrops within the range of D∼D+ dD (mm) in a unit volume of rain medium, N(D) is the raindrop size distribution, following the Marshall–Palmer negative exponential distribution (M-P distribution), and $\delta_{b}(D)$ is the raindrop backscatter cross-section (m^2^).
(16)$$\eta_{rain}=\int_{0}^{D_{\max\;}}\delta_{b}(D)N(D)dD,$$

In the loss caused by snow (17), denoted as $\gamma_{snow}$ (dB/km), I represents the snow intensity, measured in mm/h. The calculation method for the backscatter cross-section $\eta_{snow}$ of snow particles per unit volume is similar to that of rain.
(17)$$\gamma_{snow}=7.47\times 10^{-5}f\cdot I\cdot (1+5.77\times 10^{-5}f^{3}\cdot I^{0.6}),$$

The loss caused by fog $\gamma_{fog}$ (dB/km) is represented by the formula (18), where $V_{m}^{\;}$ represents visibility, measured in meters. Fog scatters electromagnetic waves, resulting in fog clutter, and the calculation formula for the backscatter cross-section $\eta_{fog}$ of fog particles per unit volume is similar to that for rain and snow.
(18)$$\gamma_{fog}=\frac{0.148f^{2}}{V_{m}^{1.43}}\;,$$

The power spectrum of ground clutter can be obtained using (19), where $\sigma_{f}^{\;}$ represents the frequency statistical standard deviation of the ground clutter power spectrum, and $f_{d}$ represents the average Doppler frequency shift of the ground clutter power spectrum, primarily influenced by the speed of the radar-equipped vehicle’s motion.
(19)$$S(f)=exp(-\frac{{\left(f-f_{d}\right)}^{2}}{2\sigma_{f}^{2}})$$

#### 4.1.4. Function Modeling Method Based on Simplified Mechanisms

Currently, the modeling approach for radar functional modules follows the working mechanism of each module. Sub-modules are established separately to simulate the functions of various components and, then, linked together according to the workflow to form an integrated radar functional model. However, during the modeling process, the sub-modules related to radar RF functions are generally simplified to some extent, primarily concentrating on key internal components of the radar, such as the waveform generator, VCO, and transmitting antenna in the transmitter, as well as the receiving antenna, mixer, and ADC in the receiver. Typically, fewer interference factors are considered for radar functional modules. The modeling content usually includes the following:(1)Signal generation. The signal source model in the transmitter generates ideal FMCW signals based on preset waveforms.(2)Signal transmission. Adjust the transmission signal power in various directions according to the directional diagram of the transmitting antenna array.(3)Signal reception. Adjust the received signal power in various directions according to the directional diagram of the receiving antenna array.(4)Signal mixing. In the mixer model, the signal generated by the signal source is used as the local oscillator signal, mixed with the frequency of the echo signal to obtain intermediate frequency signals.(5)Adding thermal noise. To simulate interference within the functional module, a thermal noise module conforming to Gaussian distribution is usually added to simulate signals more realistically.(6)Signal analog-to-digital conversion. Within the ADC, analog signals are converted to digital signals based on the preset sampling rate.

#### 4.1.5. Generic Radar Algorithms Modeling Method

The radar algorithm module uses 3D-FFT to extract the target’s distance, velocity, and angle data from raw data, outputting RAmap and RDmap. Three-dimensional-FFT includes three steps: (1)Range dimension FFT. Process the raw data in time series. Then, perform Fourier transform on the signal at each time point using the one-dimensional FFT algorithm, converting the signal from the time domain to the frequency domain. In the frequency domain, different frequency components correspond to different target distances.(2)Doppler dimension FFT. Further conduct Doppler dimension FFT processing on the frequency components of each distance unit, converting the signal from the frequency domain to the velocity domain, thus extracting the target’s velocity information.(3)Angle dimension FFT. Conduct FFT processing on the target signal determined by range and velocity in the angle dimension, converting the signal from the velocity domain to the angle domain, thereby determining the target’s azimuthal position.

The radar algorithm module commonly uses CA-CFAR to further process RAmap and RDmap, extracting point clouds of the targets from them [38,39,40]. The steps are as follows: (1)Background noise estimation. Select a set of reference cells around the cells to be detected (CUT) in the input echo map and calculate the average power of the reference cells as the estimate of background noise.(2)Compute thresholds. Then, calculate detection thresholds based on the estimated noise level and predetermined false alarm rate.(3)Target detection. If the power of CUT exceeds the threshold, it is considered to detect a target; otherwise, it is considered no target.

The radar algorithm module employs DBSCAN to cluster point clouds into targets, which operates by analyzing the density of data points [41,42,43,44]. Its fundamental steps include the following:(1)Identifying core points. It sets the distance range ε and the minimum number of neighbors MinPts, then searches for core points.(2)Identifying boundary points. Points that are not core points but fall within the ε-neighborhood of core points are labeled as boundary points.(3)Cluster expansion. Points around core points that meet the criteria are merged into a cluster, forming a target.

The radar algorithm module often utilizes methods such as Kalman filtering for target tracking. Its basic steps comprise the following:(1)Prediction phase. Utilizing previous state estimates and system models to predict the current state and updating the uncertainty of the predicted state.(2)Update phase: Using new measurement data to correct or update the predicted state calculating the Kalman gain, updating the error covariance matrix to reflect new estimation errors.

### 4.2. Modeling Methods for Signal-Level and Object-Level Radar Models

The signal-level radar model includes radar functional modules, echo modules, and 3D-FFT modules, apart from radar algorithms. The commonly used modeling method for signal-level radar models is based on mechanistic processes, combining echo models, functional models, and 3D-FFT modules. As depicted in Figure 6, this includes combinations of echo models composed of ray tracing, functional models based on simplified mechanisms, and 3D-FFT modules, as well as combinations of echo models constructed with ideal mechanisms, functional models based on simplified mechanisms, and 3D-FFT modules. Additionally, there are data-driven end-to-end modeling methods for signal-level radar models.

For target-level radar models, one modeling method is to add radar algorithm modules to the physical-process-based signal-level model to construct a complete radar model capable of outputting target-level data. However, the more common method is end-to-end modeling. There are two types of end-to-end methods, with the most common being the geometric method. This method judges occlusion based on the relative position between the target and the radar, then directly outputs unoccluded targets, while adding noise to the output target parameters to simulate interference in the radar. Another end-to-end method is data-driven modeling.

#### 4.2.1. Signal-Level Modeling Method including Echo Models Based on Ideal Mechanisms

The most commonly used signal-level radar modeling method currently is to construct an echo model based on ideal mechanisms, then build a functional model using simplified mechanisms, and finally use 3D-FFT to process data to generate RAmap and RDmap. There is a significant amount of research on this method. Preeti S. Pillai and others have utilized MATLAB’s phased array and autonomous driving toolboxes to create simplified radar functional models and ideal echo models in 2017 [45]. Huan Lei Chen applied this method to develop a functional and echo model for a 24 G radar [46]. Bing Zhu created an ideal echo model through this methodology [47]. Seok Kim, Huanlei Chen, Shuqing Zeng, among others, also constructed functional and echo radar models using this method [48,49,50]. The model constructed by this method can quickly obtain signal-level outputs, but the functional models and echo models built in the above studies are very idealized, resulting in a large discrepancy between the output signals and the signals from real radars. On this basis, some related improvement methods have been proposed to enhance the accuracy of echo models and functional models.

Improvement methods for echo models focus on enhancing the accuracy of echo signals and incorporating the effects of weather, ground, etc., on echo signals. In terms of improving the accuracy of echo signals, consideration is given to the influence of target shape on reflected signals by incorporating the reflected signals of electromagnetic waves at different positions of the target into the echo model. The most common method is to treat the target as multiple point scattering centers. For instance, Jiao Guo, Lizette Lorraine et al. selected characteristic locations on the target as scatter centers, thereafter calculating the radar’s echo signals using the target’s parameters and radar detection equations [51,52]. To improve the representativeness of scatter centers, Rajan Bhalla and Karin Schuler independently applied the bouncing ray method and the greedy algorithm to choose the most crucial locations on the target as scatter centers [53,54]. Building on this, Zoltan Ferenc Magosi further segmented the scene into multiple resolution units, considering the shape of the targets. He calculated the echo signals reflected back to the radar from each resolution unit, using the electromagnetic wave propagation formula [55]. Regarding the influence of weather and ground, Jiao Guo, Zora Slavik incorporated weather factors like rain attenuation into the echo model [51,56]. Lizette Lorraine and colleagues considered the impact of rough ground scattering on echo signals [52]. Xin Li has extensively studied the distribution characteristics of ground clutter, rain, snow, fog, and other weather clutter in different scenarios and fitted weather clutter and ground clutter under various conditions based on Gaussian distributions [57]. The aforementioned methods have significantly enhanced the computational accuracy of the echo model by improving both the propagation and scattering processes of electromagnetic waves. Unfortunately, the excessive simplifications in the modeling process mean that the improved modeling methods still yield echo models of relatively low accuracy.

Improvement methods for functional models in radar models focus on considering more functionalities of components and simulating non-ideal factors in components. Improvements to functional models mainly concentrate on consider more components and enhancing the simulation of non-ideal factors within these components. For instance, Ivan Kravchenko, Jithin Kannanthara, Weiwen Deng, and Zoltan Ferenc Magosi have developed more comprehensive radar functional models based on radar mechanisms. This model encompasses most radar functional components and introduces Gaussian-distributed thermal noise and phase noise to simulate the interference from non-ideal factors within radar functional modules [27,30,31,32,33]. Furthermore, Camilla Kärnfelt significantly enhanced the functional model’s accuracy by directly obtaining detailed parameters of certain components in the radar functional modules from suppliers [58]. Ali Bazzi, Manuel Dudek, and others employed the simulation software ADS in early 2011 to 2012 to develop functional models, thereby achieving high-precision modeling of the functional components [59,60,61]. These improvements have enhanced the models’ accuracy, particularly the models constructed by Camilla Kärnfelt with real parameters and those by Ali Bazzi and Manuel Dudek using ADS. However, the functional models still lack a comprehensive consideration of non-ideal factors in radar components, such as the dynamic changes in component parameters with frequency and power. Additionally, acquiring the real parameters of radar components is challenging, and simulations based on ADS involve a substantial computational burden.

From a modeling perspective, the combination of the echo model based on ideal mechanisms with the simplified functional model and 3D-FFT module significantly reduces computational load by avoiding the calculation of electromagnetic wave propagation in space. This approach allows for rapid simulation of the radar detection process, is unrestricted by the scene, has broad applicability, and offers high model openness. However, the simulation accuracy of this method is currently low, rendering it unsuitable for high-precision applications.

#### 4.2.2. Signal-Level Modeling Method including Echo Models Based on Ray Tracing

Replacing the echo model constructed via ideal mechanisms, the ray-tracing-based echo model, combined with the functional model and 3D-FFT module, offers a more detailed account of electromagnetic wave propagation in space. This approach can capture the impact of target shape, material, ground medium, and multipath propagation of electromagnetic waves on the radar echo signal, and it also mirrors the radar’s signal processing sequence. Since the previous section already introduced the modeling method and improvements of the functional model, this section will focus on the application of the ray tracing-based echo modeling method. In current research, Manuel Dudek, Martin Holder, Xiyu Wang, and Stefan O. Wald have independently developed radar echo models using ray tracing methods [60,61,62,63,64]. The echo model constructed by this method can obtain relatively accurate echo signals, but it comes with a huge amount of computation, consuming a lot of computing resources and time. In response to this situation, some methods to improve computational speed have been proposed.

Since the ray tracing model consumes a lot of computational resources in the ray pathfinding and the far-field radiation calculation on the target surface, improvements to the computing speed of the ray tracing method are mainly focused on these two steps. Marco Ciarambino introduced an echo calculation approach utilizing Unreal Engine on a graphics processing unit (GPU) [65], significantly enhancing the model’s computational speed through parallel processing. Yet, Unreal Engine’s ray tracing technique is not fully adaptable to millimeter-wave frequencies, producing echo signals that align more closely with optical frequencies and large discrepancies with real radar echoes. Kazem Sabet pre-established databases for ray transmission paths and electromagnetic wave scattering on target surfaces. Using these databases, he constructed channel transfer functions and retrieved target echo information directly [66]. This approach significantly increases the echo model’s computational speed, but its reliance on database content, making it inapplicable to dynamic autonomous driving scenarios. Leire Azpilicueta introduced a hybrid approach combining ray tracing with neural networks [67]. Utilizing neural networks, this method predicts reflection points for rays propagating through space, conserving resources and time required for calculating ray paths. However, this technique necessitates segmenting the surrounding scene into various grids for neural network training based on the partitioned environment. When the test scene changes, this ray path prediction model will no longer be applicable. Nils Hirsenkorn and colleagues employed the geometric optics (GO) method to simulate the complete propagation of rays in space, taking into account the diffusion of rays on curved surfaces [68]. This method avoids far-field radiation calculations on the target and achieves near real-time computing speeds by reducing the number of rays and their bounces, but it results in lower computational accuracy. Ushemadzoro Chipengo and Juan D. Castro, among others, employed Ansys’s SBR+ technique to develop several echo models [69,70,71,72,73,74,75], simplifying the computational steps for rays on object surfaces in ray tracing, incorporating pre-training, and leveraging GPU computations to conserve time. Despite enhancing accuracy and significantly reducing computation time compared to traditional ray tracing, it regrettably fails to achieve real-time simulation speeds.

Radar models constructed based on ray tracing methods can be applied to various scenarios and can achieve a very high level of computational accuracy for echo signals. However, their computational load is directly proportional to the accuracy, demanding significant computing resources and time for high precision. At present, it is not possible to simultaneously fulfill the demands for high computational accuracy and speed. Additionally, inadequate functional models may lead to an increase in error in the final output of radar models.

#### 4.2.3. Signal-Level Modeling Method Based on End-to-End Approach

End-to-end signal-level modeling typically employs machine learning techniques to develop radar models. Scene information is chosen based on the input and output requirements of the signal-level model, with radar output echo maps as the model’s output. Input and output are often represented in image form, with corresponding network models such as convolutional neural network (CNN) and a mixture of Gaussians commonly used to train radar models to fit the mapping relationship between input and output images, facilitating the processing of image information. Since such models include the entire process of the radar echo module, functional module, and algorithm module, their model structure is usually complex and highly nonlinear. Moreover, this approach necessitates extensive data for training and validation. Building test scenarios to establish datasets or using publicly available radar datasets, such as ColoRadar, RaDICaL, CARRADA, RODNet [76,77,78,79] is a common approach. In relevant studies, Tim A. Wheeler introduced a modeling method that utilizes a conditional probability distribution model, based on scene information, to predict the radar’s RAmap. The input data encompass a list of objects that describe dynamic targets, like vehicles and pedestrians, and a spatial raster detailing the ground’s fixed elements, such as roads and grass. Utilizing convolutional and fully connected layers, the model integrates these two modalities of scene information into scene feature variables. The model leverages a Conditional Variational Autoencoder (CVAE) to ascertain the conditional probability distribution parameters linking these features with the RAmap. Adversarial networks are employed during training to augment the authenticity of the generated data. Upon completion of training, a noise module is integrated into the model generation process [80]. This method facilitates the prediction of the radar’s RAmap; however, the model’s accuracy is moderate and falls short of precisely forecasting the radar output echo map.

Data-driven end-to-end signal-level modeling methods cannot access the internal signals of radar models, but they can achieve real-time computational speed. Despite their potential to achieve high computational accuracy, current models have yet to demonstrate this capability. This deficiency primarily stems from the scarcity of research in this domain and the absence of high-quality datasets. The construction of radar models demands a substantial volume of precise data, encompassing echo images of radar outputs and accurate ground truth information regarding targets in the scene. Unfortunately, publicly available datasets often lack the necessary precision in ground truth data, significantly obstructing the advancement of high-precision signal-level data-driven radar models. Additionally, these data-driven models grapple with challenges related to generalization. 

#### 4.2.4. Object-Level Modeling Method Based on Physical Processes

The physical-process-based object-level radar modeling approach typically incorporates CFAR, clustering, and tracking algorithms into the foundational-process-based signal-level radar modeling method, enabling the extraction of object-level outputs from echo maps. Object-level radar models produce outputs that encompass the detectability and parameters of the object. In existing studies, Preeti S. Pillai, Zoltan Ferenc Magosi, Ivan Kravchenko, Jiao Guo, Lizette Lorraine, and Jithin Kannanthara have all incorporated CFAR and clustering modules into their signal-level models to extract lists of targets [51,52,55,81,82].

The characteristics of this method are similar to those of corresponding signal-level radar models, with wide applicability and high openness. However, the radar algorithm modules added to the signal-level model by this method introduce additional errors, resulting in poor prediction ability for the false alarm and missed alarm states of radar output objects and object parameters.

#### 4.2.5. Object-Level Modeling Method Based on Geometric Method

The geometric method does not consider the mechanism of radar detection. The radar is conceptualized as emitting a cone from a vertex at the center of the radar in the direction of the radar’s normal, with the cone’s size determined by the radar’s effective detection range. This method, based on the scattering center approach, selects typical positions on the target surface as feature points. It determines target detectability by calculating whether these feature points are illuminated by the cone. The method proceeds as follows: (1)Determination of effective detection range. Calculate whether the target is within the radar’s detection range based on the geometric relationship between the target and the radar. Feature points within the detection range are considered identifiable.(2)Occlusion assessment. Calculate the occlusion relationship between targets based on their positions. Feature points not occluded are considered detected.(3)Target identification. Determine whether the target is recognized based on the number of identified feature points on the target surface, and output the true parameters of the recognized targets.(4)Introducing random error. Gaussian noise is added to the true parameters of the output targets to simulate detection errors resulting from various interferences during radar detection.

In current research, M. Stolz regards targets as point targets, assessing their detectability based on the radar’s detection scope and the target’s location, subsequently outputting the true parameters of detected targets [83]. T. Hanke and others also view targets as point targets, adding Gaussian noise to the scene’s true values to simulate detection errors as the model’s output [84]. Xin Li delved into the mechanisms of error generation in radar ranging, velocity measurement, and angle measurement, adding error correction terms to the scene’s true values to achieve higher accuracy [85]. In studies on target detectability, Stefan Muckenhuber and others view the target as a bounding box, construct the radar’s field of view (FOV) area, perform projection calculations on targets within the area to determine occlusion relationships, and, thus, judge detectability. They also statistically analyze the probability of false alarms and missed alarms from real data, incorporating this probability into the calculation of target detectability to improve the accuracy of false and missed alarm detection [86]. Bing Zhu and colleagues have meticulously analyzed how occlusion and the radar’s detection scope affect target detectability [47]. Zhan Jun accounted for the RCS values of targets in various orientations, inter-target occlusion relationships, and the impact of the radar detection range on target echo energy. Furthermore, he contrasted the intensity of the target echoes received by the radar with the detection threshold to ascertain target detectability [87]. Xin Li mechanistically analyzed the genesis of radar false and missed alarms, formulating probability functions for these alarms to deduce target detectability [85]. It is apparent that current geometric approaches in developing object-level models prioritize the assessability of target detectability, with the simulation of target detection errors confined to the addition of Gaussian noise, yielding restricted accuracy. Moreover, contemporary methods for determining target detectability fail to accurately predict outcomes of false and missed alarms, particularly manifesting substantial inaccuracies in false alarm predictions.

Geometric methods enable rapid acquisition of radar detection results and are applicable to any scenario. While this method demonstrates some accuracy in missed alarm prediction, the prediction of radar object errors mostly relies on adding Gaussian noise. Furthermore, predictions for false alarm objects are relatively straightforward, leading to generally low accuracy in predicting false alarm objects and object parameters.

#### 4.2.6. Object-Level Modeling Method Based on Data-Driven Approach

Data-driven object-level end-to-end modeling methods select the state parameters of targets in the scene as input and the object list outputted by the radar as output. They use machine learning models to establish the mapping relationship between input and output. Since this model’s inputs and outputs use parameter lists, the amount of information contained in the input parameters may be limited. Careful selection of scene parameters that significantly affect radar detection results is needed to reduce the omission of key information. This model employs various modeling techniques, including artificial neural networks, support vector machines, and Gaussian mixture algorithms [88,89,90,91]. Targets in the radar working scenario may change, and the object list outputted by the radar may also change due to false alarms and missed alarms. These changes result in a lack of stability in the number and order of parameters in the model’s input and output. The key to this modeling method lies in maintaining the stability of the mapping relationship between the scene targets in the model’s input and the radar objects in the output, despite changes in the state or order of the input targets and output objects.

Current data-driven models primarily focus on modeling errors in radar output target parameters. Philip Aust introduced a Gaussian mixture radar modeling method to determine the position distribution on the target surface of radar-detected targets [92]. By conducting multiple measurements, he acquired several sets of radar output target data within the same scene, aggregated these datasets to reveal the distribution of targets surrounding the target vehicle, and subsequently modeled this distribution using a mixture of Gaussians. This approach does not provide predictions for the target’s speed and RCS parameters. Moreover, given the simplicity of the scenarios, the models established via this method exhibit very poor generalizability. Alexander Suhre utilized a CVAE, inputting the true value parameters of scene targets to predict errors in radar detection outcomes. Models developed with this method can anticipate trends in target changes, albeit with substantial errors [93]. Alexander Scheel applied a variational Gaussian mixture technique to compute the distance, velocity, and angular values of radar outputs [94]. The model inputs the vehicle’s state vector, encompassing kinematic state and dimensions, estimating these parameters via Bayesian methods. This model yields commendable predictive outcomes in straightforward scenarios but has not been validated within complex scenes. Furthermore, it cannot predict the false alarm and missed alarm statuses of targets. Hexuan Li employed a mixture of Gaussians to model the velocity, range, and angular inaccuracies found in radar detection results [95]. However, diverging from standard Gaussian mixtures, he utilizes a multilayer perceptron to model the means, variances, and Gaussian mixture coefficients within the Gaussian mixture model. This method can achieve predictions of the error probability distribution, but there are significant errors in speed and angle predictions, and it lacks predictions for target RCS. Takashi Owaki and others proposed a data-driven and ray tracing hybrid simulation method for target RCS [96]. Target information in the scene is represented using normal maps, depth maps, position maps, etc. A CNN network predicts the signal strength distribution on the target surface, creating an intensity map. Ray tracing calculates the phase distribution on the target surface, generating a phase map. Then, intensity maps, normal maps, and depth maps are used as inputs to train the target’s RCS values with second CNN. The complex structure of this model brings high computational accuracy, but the calculation speed is slow.

In terms of predicting false alarms and missed alarms for targets, N. Hirsenkorn conducted research on the phenomena of false alarms and missed alarms in radar detection results, proposing a method based on kernel density estimation to construct probability density functions to predict radar detection outcomes. This model uses the true range of target in the scene as input and the range of object detected by radar as output. The model can predict the false alarm and missed alarm status of targets and range errors, but it shows lower accuracy. Stefan Muckenhuber proposed four different radar models for judging the target detection characteristics [97]. The first one directly judges the identification characteristics of a target by setting an RCS threshold and based on the RCS value of the target’s echo. The other three are data-driven radar models that use support vector classifier, Random Forest, and Gradient Boost methods to process the true values of the scene and use them as model inputs, directly predicting target detectability through deep learning models. Among them, the data-driven models showed better classification effects. Among them, the data-driven models showed better classification effects.

Data-driven end-to-end object-level modeling methods directly compute the mapping relationship between targets in the scene and radar output objects, demonstrating very fast computation speed and a certain level of accuracy in predicting object parameters and false alarms and missed alarm states. This method requires a large amount of object-level radar data and the corresponding ground truth of the scene for training. However, current publicly available radar datasets such as Oxford Radar RobotCar, nuScenes, MulRan, and RadarScenes usually lack ground truth of scene targets [98,99,100,101], necessitating data collection by model builders to establish datasets. Existing data collection efforts often focus on relatively simple scenarios where occurrences of missed alarms and false alarms are minimal. Furthermore, the current method does not fully consider the principles of radar detection and the generation of detection errors, and the selected model input parameters may not comprehensively reflect the influence of the environment on radar detection results. The simplicity of the model structure also partially limits the computational capability of such methods. Lastly, these methods also face challenges in generalization.

### 4.3. Summary of Modeling Methods

As shown in Table 2, for modeling methods that construct echo models based on ideal mechanisms, although they can quickly predict the echo signals of targets, their overly low echo accuracy severely affects the prediction outcomes. For modeling methods that construct echo models based on ray tracing, although ray tracing can achieve high precision calculations of the echo module, an inaccurate functional model will reduce the overall modeling effect. Furthermore, the computational efficiency of ray tracing is also a key factor affecting model performance. Data-driven signal-level modeling methods can meet the requirements for computation speed and accuracy in some special scenarios, but the current methods have limited analysis on radar detection mechanisms, and there is significant room for improvement in accuracy prediction. They also face issues with generalizability.

In object-level modeling methods, as shown in Table 3, methods based on physical processes not only have the problems present in signal modeling methods but also suffer from poor radar algorithm model performance, which greatly restricts the application of process-based methods in object-level modeling. Geometric modeling methods are overly simplified, and their prediction accuracy cannot meet the requirements of most applications. Data-driven methods feature fast computation speeds and have demonstrated feasibility in predicting radar parameters and target states, and although the current accuracy has not yet achieved high precision, there is considerable development potential. However, these methods also face issues with generalizability.

## 5. The Current Challenges and Development Directions of Radar Modeling Methods

As autonomous driving technology continues to advance to higher levels, the requirements for radar model performance in various applications are constantly increasing. In the future, radar algorithms and sensor fusion algorithms will strengthen data processing and optimization to improve perception accuracy and enhance adaptability to different environments. To support higher-precision autonomous driving decision-making and planning, standards for radar selection, installation, and testing also need to be correspondingly improved. Additionally, data augmentation techniques will also increase the demand for data accuracy to establish higher-quality datasets. In the development of radar RF hardware, besides pursuing higher simulation accuracy, there will also be an increased demand for openness of radar models to more accurately test the performance of radar RF components.

Faced with these challenges, existing signal-level radar modeling methods, including ideal-mechanism-based echo modeling methods and simplified-mechanism-based functional modeling methods, struggle to achieve the required high-precision computational results. The future development of signal-level models will focus on three key directions: firstly, the development of high-precision and high-openness radar functional modeling methods based on microwave RF mechanisms; secondly, improving the efficiency of ray tracing modeling methods; and thirdly, enhancing the accuracy and generalization of end-to-end data-driven models. Among these, developing high-precision and high-openness radar functional models will require deeper research and finer modeling of radar RF principles. The efficiency improvement of ray tracing modeling methods will depend on the progress of high-performance computing devices such as GPUs, and improvements in artificial intelligence algorithms will also drive efficiency improvements. In terms of data-driven modeling, with the rapid development of artificial intelligence technology, such as the application of large-scale models and the continuous expansion of datasets, it is expected that the performance of data-driven models will continue to improve, meeting the comprehensive requirements of simulation testing for speed, accuracy, and generalization in the future.

For object-level modeling methods, whether based on mechanism-based or geometric methods, achieving higher precision poses significant challenges. The development of object-level modeling methods increasingly relies on the advancement of data-driven modeling methods. By delving into detection mechanisms and utilizing higher-performance model structures and higher-quality datasets, the problem of low model accuracy can be addressed. Attention should also be paid to methods for improving model generalization performance.

## 6. Conclusions

Radar modeling methods play a critical role in the development of autonomous driving. To better apply radar models to promote the development of autonomous driving and guide the high-quality development of radar modeling methods, it is necessary to comprehensively consider the key aspects and interference factors in the radar working process and to clearly evaluate the key features of modeling methods based on the actual application requirements. A thorough assessment should be conducted based on the principles of existing radar modeling methods. This can help determine the characteristics of different modeling methods to facilitate the construction of corresponding radar models according to actual application needs. Furthermore, the development direction of radar modeling methods should be determined based on the direction of autonomous driving development.

This paper first clarifies the content of radar modeling by analyzing the radar detection mechanism to identify the contents of the RF functional module, echo module, and algorithm module in the radar working process, as well as the existing interference factors. Then, it analyzes the differences in the requirements of various applications in autonomous driving for radar models, determining four indicators for evaluating radar models: accuracy, speed, openness, and generalization. Subsequently, the paper comprehensively analyzes the modeling methods for each module in the radar and the overall modeling methods for target-level radar models and signal-level radar models composed of various module modeling methods. It introduces the research status of overall radar modeling methods and improvements made by researchers based on evaluation criteria to clarify the advantages and disadvantages of each modeling method. Finally, based on the demands of autonomous driving for radar models, the future development directions of radar modeling methods are proposed. This study clearly defines the requirements of various application scenarios for radar models and the effects that various modeling methods can achieve and also delineates future development directions, thereby promoting the application of radar modeling methods and the future development of radar modeling methods in autonomous driving.

## Figures and Tables

**Figure 1 sensors-24-03310-f001:**
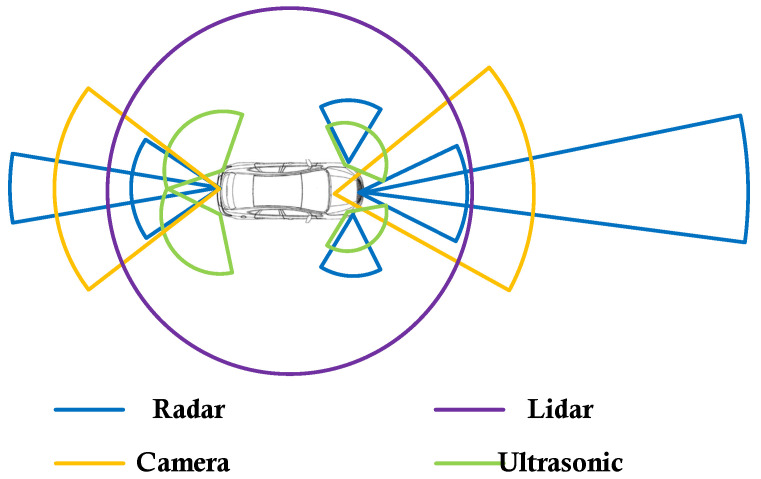
Autonomous driving sensors.

**Figure 2 sensors-24-03310-f002:**
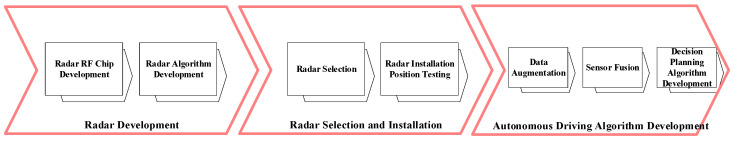
Application of radar model in autonomous driving.

**Figure 3 sensors-24-03310-f003:**
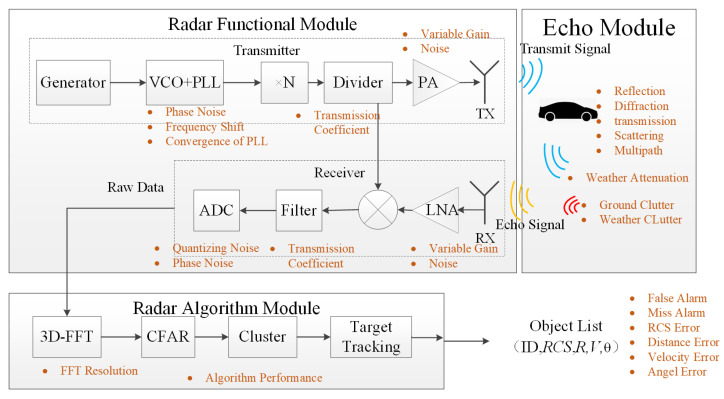
Radar system and radar workflow.

**Figure 4 sensors-24-03310-f004:**
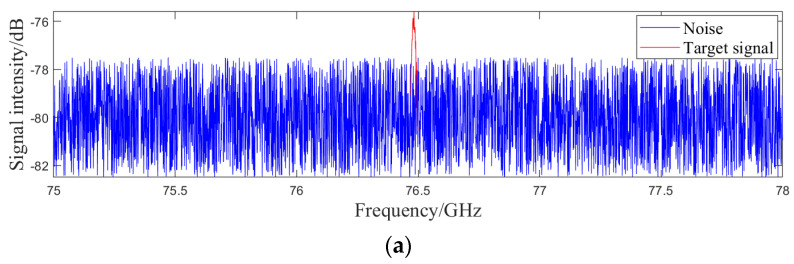
Radar raw signal containing interference signals. (**a**) The target is identified (**b**) The target is not identified.

**Figure 5 sensors-24-03310-f005:**
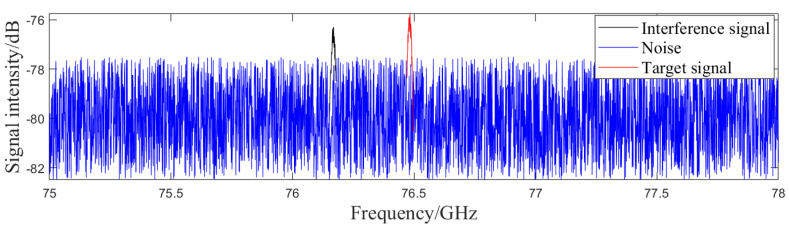
Radar raw signal containing multipath signals.

**Figure 6 sensors-24-03310-f006:**
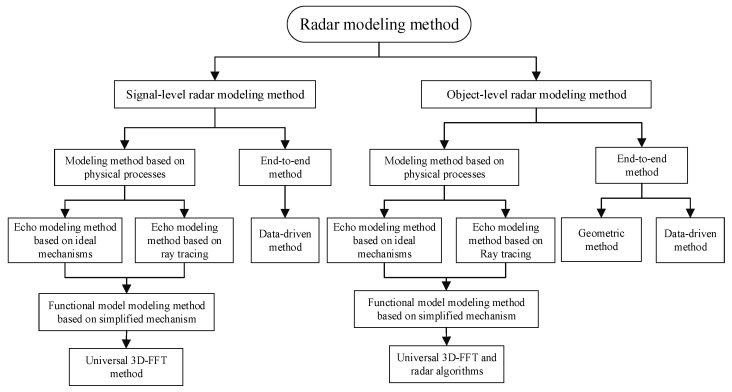
Radar modeling method.

**Table 1 sensors-24-03310-t001:** Levels of demand for radar models in different applications.

Characteristics	Decision Planning Algorithm Development	Sensor Fusion		Data Augmentation		Radar Algorithm Development	Radar RF Hardware Development	Radar Installation Position Simulation	Radar Selection
		Target Level	Signal Level	Target Level	Signal Level				
Accuracy	High	High	High	Medium	Medium	High	High	Medium	Medium
Speed	High	High	High	Low	Low	Low	Low	Low	Low
Openness	Low	Low	High	Low	High	High	High	High	Low
Generality	High	High	High	High	High	High	Low	Low	Low

**Table 2 sensors-24-03310-t002:** Performance evaluation of signal-level modeling methods.

Characteristics	Echo Modeling Method Based on Ideal Mechanisms	Echo Modeling Method Based on Ray Tracing	Data-Driven Method
	Functional Model Modeling Method Based on Simplified Mechanism		
	3D-FFT		
Accuracy	Low	Medium	Medium
Speed	High	Low	High
Openness	High	High	High
Generality	High	High	Low

**Table 3 sensors-24-03310-t003:** Performance evaluation of object-level modeling methods.

Characteristics of Models	Echo Modeling Method Based on Ideal Mechanisms	Echo Modeling Method Based on Ray Tracing	Geometric Method	Data-Driven Method
	Functional Model Modeling Method Based on Simplified Mechanism			
	3D-FFT and Radar Algorithms			
Accuracy	Low	Medium	Low	Medium
Speed	High	Low	High	High
Openness	High	High	Low	Low
Generality	High	High	High	Low

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
