# Peer review of "An Overview of Millimeter-Wave Radar Modeling Methods for Autonomous Driving Simulation Applications"

_sensors, 2024, doi:10.3390/s24113310_

Round 1

Reviewer 1 Report

Comments and Suggestions for Authors

To be honest, the manuscript is not well organized and written. I would like to suggest a major revision.

I would like to suggest shortening Section 2. The principle of the radar in the manuscript is textbook content. It’s unnecessary to introduce the principle of radar measurement about range, velocity, and DOA with large paragraphs in a journal paper. It’s nothing to do with autonomous driving.

One of the key parts of the radar for the autonomous driving simulation is the radar echoes under realistic environments. The disadvantages such as low resolution, sparsity, clutter, high uncertainty, lack of good datasets, and so on, have limited radar for the autonomous application. However, the manuscript lacks discussions on the resolution, sparsity, and clutter suppression of the radar modeling.

Comments on the Quality of English Language

There are many errors in the manuscript. It's impossible to list all of them. I strongly suggest the authors carefully read through the manuscript and correct the errors. For example,

Line 50, Line 62, Line 64, there are obvious typos. 

All the subtitles in Section 2 are misnumbered. 

Author Response

We are grateful for the opportunity to revise our manuscript, " An Overview of Millimeter-Wave Radar Modeling Methods for Autonomous Driving Simulation Applications". We truly appreciate the time and effort you invested in reviewing our work. Based on your suggestions, we have made significant revisions to the manuscript:

Content Adjustment: We have addressed your concern regarding the redundancy of radar principles in Section 2. Specifically, we have removed extensive discussions on radar detection principles as they are commonly known and may not directly contribute to the topic of radar modeling for autonomous driving. Instead, we have enriched the section by providing a detailed exploration of various interference factors encountered in radar operations, such as weather clutter, terrain clutter, multipath effects, radar resolution limitations, deficiencies in radar algorithms, and the impact of sparse point clouds.

Enhanced Evaluation Metrics: We have delved deeper into the description of radar modeling evaluation metrics in Section 3, providing a more thorough and substantiated discussion to enhance its clarity and coherence.

Improved Methodology Description: We have revamped the presentation of radar modeling methods in Section 4, ensuring a clearer delineation of the principles and characteristics of various modeling techniques. We have included discussions on corresponding modeling approaches about weather clutter and terrain clutter.

Refined Analysis in Later Sections: Sections 5 and 6 have undergone thorough revisions to provide more insightful analyses and logical coherence.

Comprehensive Review for Errors and Abbreviations: We have conducted a comprehensive review to rectify any inaccuracies, errors, or inconsistencies in language usage and abbreviations throughout the manuscript.

Thank you again for your thorough and valuable reviews, and constructive suggestions, which are very helpful for us to improve the paper quality.

Yours sincerely,

Kaibo Huang and Co-authors

Reviewer 2 Report

Comments and Suggestions for Authors

This manuscript is devoted to review of various modern radar modeling methods,  identification of  key radar modeling elements, assessing current radar modeling technologies, and projecting future trends in the field of radar models based on the development direction of autonomous driving.  Authors carried out an overview of the principles and characteristics of various radar modeling techniques, summarizing their merits and limitations.  The topic of manuscript is important for scientific and technical groups in areas of development of radar modeling and autonomous driving.

There are some points to make the information more clear or to correct some details:

1)      It is necessary to explain the abbreviations at first mention. There are the abbreviation and without explanation (e.g. RF for radiofrequency (first mention is at 62nd line), AD for analog-to-digital conversion (first mention is at 128th line); DBSCAN (first mention is at 128th line), GO for geometric optics possibly (560th line); GPU (546th line); FOV for field of view (649th line) ).  The abbreviation for “three-dimensional FFT (Fast Fourier Transform Algorithm)” (130th line) should be “3D-FFT” as in Figure 3. The first mention for Cell Averaging CFAR (CA-CFAR) should be in 360th line but not in 428th as now.

It is necessary to explain all abbreviations and values appeared in Figure in Figure captions or in the text before Figure (e.g., the Figure 3 contains a large number of abbreviations (VCO, PLL, PA, LNA, 3D-FFT etc.) and values (R,q, V, s etc.) that did not appear before this Figure and the Figure  caption does not contain their interpretations).

2)      There is the caption in Figure 2 “Data Augmentation” in block of “Autonomous Driving Algorithm Development”. Does it mean “Data Accumulation”? It is necessary to check there and further.

3)      Besides absence of decoding the abbreviations and values, the echo signal in Figure 3 is noted but the probing signal is unsigned.

4)      The parameter of f0 in Figure 4 is not explained.

5)      The values in formulas should be checked and explained after formula. The Δ𝑓𝑑, T  are not explained after (1). Authors say about time delay, but there is no this value in text (reader will have to guess that Δt in figure (5) is this delay). It can be introduce in 152nd line at first mention of one. In (2) the value of velocity is v but target's radial velocity 𝑉�� is appeared in the text before.

6)      What is “A” (Y-axis) in Figure 6?

7)      What are “RX1” and “RX2” in Figure 7? If the authors introduce axes in Fig. 7, then they need to be defined somehow.

8)      Authors use the term of “application” for “simulation accuracy, speed, generalizability, and the level of internal openness” (325th-326th lines). These values are characteristics of model but not applications. Besides authors write about “levels of demand for radar models in different applications” (Table 1) and use terms “high”, “medium” and “low”, but how do these terms differ from each other? How are they differentiated? It is not clear.

9)      The Tables 2 and 3 should have the common line over three right columns with title of “Modeling methods” and the title of first column should be “Characteristics of models” or something else. Besides there are no any numerical estimations of these characteristics for modeling methods discussed in manuscripts.

10)  There are some typos in text. Some examples of typos are further. It  is written “…perception, As depicted in Figure 1.” (50th line), it is necessary “…perception, as depicted in Figure 1.”. The words after “;” should be written with lowercases (e.g., “algorithms; Autonomous” (62nd line); “locations; Algorithm” (64th line)). It is written “Radar algorithm And DSP” instead “and” in Figures 3 and 8. It is written “chrips” instead of “chirps” in Figure 4.

This manuscript is written sufficiently clear and describes with details the results of overview of modern Radar modeling methods.           .

The manuscript can be published after minor revisions.

Comments on the Quality of English Language

 Authors should check the text for terms and correct the typos.

Author Response

Dear Editor and Reviewer ,
We are grateful for the opportunity to revise our manuscript, " An Overview of Millimeter-Wave Radar Modeling Methods for Autonomous Driving Simulation Applications". We appreciate the constructive feedback provided, which has been instrumental in enhancing the quality and clarity of our work. Before answering your question, there is a situation that needs to be clarified. Regarding the feedback from another reviewer, we have made significant adjustments to the manuscript, including:
1.Removal of detailed radar detection principles and addition of discussions on various interference factors in Section 2, such as weather clutter, terrain clutter, and multipath effects.
2.In-depth exploration of radar modeling evaluation metrics in Section 3 with more substantiated descriptions.
3.Improved presentation of radar modeling methods in Section 4, clarifying the principles and characteristics of various modeling techniques, and inclusion of discussions on modeling approaches about weather clutter and ground clutter.
4.Enhanced analysis in Sections 5 and 6 for improved logical coherence.
Due to these significant adjustments, some of the content raised in your feedback has been modified. We will now detail the specific modifications made in response to your comments.
Reply to the comments of Reviewers
Comments 1: It is necessary to explain the abbreviations at first mention. There are the abbreviation and without explanation (e.g. RF for radiofrequency (first mention is at 62nd line), AD for analog-to-digital conversion (first mention is at 128th line); DBSCAN (first mention is at 128thline), GO for geometric optics possibly (560th line); GPU (546th line); FOV for field of view (649th line) ). The abbreviation for “three-dimensional FFT (Fast Fourier Transform Algorithm)” (130th line) should be “3D-FFT” as in Figure 3. The first mention for Cell Averaging CFAR (CA-CFAR) should be in 360th line but not in 428th as now.
It is necessary to explain all abbreviations and values appeared in Figure in Figure captions or in the text before Figure (e.g., the Figure 3 contains a large number of abbreviations (VCO, PLL, PA, LNA, 3D-FFT etc.) and values (R,q, V, s etc.) that did not appear before this Figure and the Figure  caption does not contain their interpretations).
Reply: Thank you for your comments. We have meticulously revised the manuscript to ensure that each abbreviation is explained upon its first occurrence in the text. Additionally, we have adjusted the placement of Figure 3 and other figures, and added explanations for terms and abbreviations within the figures to ensure that every term is clearly defined. 
Comments 2: There is the caption in Figure 2 “Data Augmentation” in block of “Autonomous Driving Algorithm Development”. Does it mean “Data Accumulation”? It is necessary to check there and further.
Reply: Thank you for your comment. In Figure 2, the term "Data Augmentation" refers to a method used to augment the volume of data required for data-driven approaches. This technique addresses the issue of insufficient data by incorporating simulated data into limited real-world datasets, thereby expanding the volume of data available for model training and testing. We have clarified this in the manuscript to avoid any misunderstanding.
Comments 3: Besides absence of decoding the abbreviations and values, the echo signal in Figure 3 is noted but the probing signal is unsigned.
Reply: Thank you for pointing out the oversight. We have now added labels for the probing signals in Figures 3 and 8 to ensure clarity.
Comments 4: The parameter of f0 in Figure 4 is not explained.
Reply: Thank you very much for your observation. In Figure 4, f_0refers to the starting frequency of the chirp. The content in question has been adjusted, and the manuscript no longer includes it.
Comments 5: The values in formulas should be checked and explained after formula. The Δ??, T are not explained after (1). Authors say about time delay, but there is no this value in text (reader will have to guess that Δt in figure (5) is this delay). It can be introduce in 152nd line at first mention of one. In (2) the value of velocity is v but target's radial velocity ? s appeared in the text before.
Reply: We greatly appreciate your valuable input. We have carefully reviewed and standardized the notation for radial velocity throughout the manuscript to avoid any inconsistencies. The Δ?? , T, Δt in question has been adjusted, and the manuscript no longer includes Figure 5 and (1), (2).
Comments 6: What is “A” (Y-axis) in Figure 6?
Reply: In Figure 6, the 'A' on the Y-axis represents the amplitude of the signal. The content in question has been adjusted, and the manuscript no longer includes Figure 6.
Comments 7: What are “RX1” and “RX2” in Figure 7? If the authors introduce axes in Fig. 7, then they need to be defined somehow.
Reply: In Figure 7, "RX1" and "RX2" represent different receiving antennas within the radar antenna array. The content in question has been adjusted, and the manuscript no longer includes Figure 7.
Comments 8: Authors use the term of “application” for “simulation accuracy, speed, generalizability, and the level of internal openness” (325th-326th lines). These values are characteristics of model but not applications. Besides authors write about “levels of demand for radar models in different applications” (Table 1) and use terms “high”, “medium” and “low”, but how do these terms differ from each other? How are they differentiated? It is not clear.
Reply: Thank you for your suggestion. In the manuscript, we have changed the term "applications" to "characteristics" in Table 1. We have also added explanations for the terms "high", "medium", and "low", and the criteria used for their differentiation, in the second paragraph of Section 3.
Comments 9: The Tables 2 and 3 should have the common line over three right columns with title of “Modeling methods” and the title of first column should be “Characteristics of models” or something else. Besides there are no any numerical estimations of these characteristics for modeling methods discussed in manuscripts.
Reply: Thank you for your feedback. We have modified the descriptions in Tables 2 and 3. Our assessment of the modeling methods is based on an analysis of the mechanics of each method and their effectiveness as reported in those papers. Due to the lack of unified and direct numerical evaluations in the referenced articles, it is challenging to establish specific numerical estimates for the characteristics of these methods. To make the evaluation more convincing, we have endeavored to include more analytical content in our assessment of the radar modeling methods to evaluate the characteristics of the models they produce.
Comments 10: There are some typos in text. Some examples of typos are further. It is written “…perception, As depicted in Figure 1.” (50th line), it is necessary “…perception, as depicted in Figure 1.”. The words after “;” should be written with lowercases (e.g., “algorithms; Autonomous” (62nd line); “locations; Algorithm” (64th line)). It is written “Radar algorithm And DSP” instead “and” in Figures 3 and 8. It is written “chrips” instead of “chirps” in Figure 4.
Reply: Thank you for pointing out these issues. We have corrected the spelling mistakes in the text and figures. Regarding the term "chirps" in Figure 4, it is not a typo; "chirps" refers to a type of linear frequency modulation signal, which is a term specific to the field of radar signal processing. 
Thank you again for your thorough and valuable reviews, and constructive suggestions, which are very helpful for us to improve the paper quality. 
Yours sincerely,
Kaibo Huang and Co-authors 
